# Mechanics and Energetics of Human Feet: A Contemporary Perspective for Understanding Mobility Impairments in Older Adults

Kota Z. Takahashi [1], Rebecca L. Krupenevich [2,*], Amy L. Lenz [3], Luke A. Kelly [4], Michael J. Rainbow [5] and Jason R. Franz [2]

1   Department of Health and Kinesiology, University of Utah, Salt Lake City, UT 84112, USA
2   Joint Department of Biomedical Engineering, University of North Carolina at Chapel Hill and North Carolina State University, Chapel Hill, NC 27599, USA
3   Department of Orthopaedics, University of Utah, Salt Lake City, UT 84108, USA
4   School of Human Movement & Nutrition Sciences, University of Queensland, Brisbane, QLD 4072, Australia
5   Department of Mechanical and Materials Engineering, Queen's University, Kingston, ON K7L 3N6, Canada
*   Correspondence: rlkrup@gmail.com

**Abstract:** Much of our current understanding of age-related declines in mobility has been aided by decades of investigations on the role of muscle–tendon units spanning major lower extremity joints (e.g., hip, knee and ankle) for powering locomotion. Yet, mechanical contributions from foot structures are often neglected. This is despite the emerging evidence of their critical importance in youthful locomotion. With the rapid growth in the field of human foot biomechanics over the last decade, our theoretical knowledge of young asymptomatic feet has transformed, from long-held views of the foot as a stiff lever and a shock absorber to that of a versatile system that can modulate mechanical power and energy output to accommodate various locomotor task demands. In this perspective review, we predict that the next set of impactful discoveries related to locomotion in older adults will emerge by integrating the novel tools and approaches that are currently transforming the field of human foot biomechanics. By illuminating the functions of the feet in older adults, we envision that future investigations will refine our mechanistic understanding of mobility deficits affecting our aging population, which may ultimately inspire targeted interventions to rejuvenate the mechanics and energetics of locomotion.

**Keywords:** elderly; gait; neuromechanics; foot; ankle; footwear

## 1. Current Understanding of the Mechanics and Energetics of Older Adults

For decades, biomechanics investigations on age-related declines in mobility have largely focused on the role of major lower extremity extensor muscles surrounding the hip, knee and ankle joints. These studies have shaped our understanding of how older adults, without the presence of other musculoskeletal pathologies, produce the mechanical power required for locomotion. For example, compared to younger adults, older adults walk with reduced ankle power generation and redistribute the muscular demands to the hip joint [1]—a strategy that may be linked to a greater whole-body metabolic energy cost [2], which may potentially accelerate fatigue or reduce walking independence. While these studies are informative, the mechanical contributions of more distal structures in the foot are often neglected. This is despite the growing evidence of the foot's importance in healthy and youthful locomotion. At least in young adults, the foot can not only act as its own power supply [3] but also function as a lever to alter the mechanics of more proximal leg muscles (e.g., ankle plantar flexors) [4]. Thus, investigations into the mechanics and energetics of locomotion among older adults that neglect the foot's contributions are fundamentally incomplete, ultimately limiting our ability to prescribe evidence-based interventions. As

we survey the current state of the art and contemporary understanding of age-related declines in locomotor functions, we predict that the next set of impactful discoveries will emerge from integrating the novel tools and approaches that are currently transforming the field of human foot biomechanics.

While many studies involving older adults have examined ankle kinetics, kinematics and some aspects of foot mechanics (e.g., kinematics, plantar pressure) [5], until recently, there were relatively few studies investigating mechanical energetics in asymptomatic feet in older adults [6,7]. During level-ground walking at a constant speed, young adults typically perform more negative work (i.e., energy absorption) than positive work (i.e., energy return or generation), thereby losing mechanical energy each step. Comparatively, feet in older adults lose even more energy [6,7], due to both a greater negative work and a reduced positive work. Moreover, this energy loss in older adults is exacerbated when walking fast or against impeding forces that challenge the propulsion [6]. Cumulatively, these findings may highlight the inability of older adults to appropriately modulate the foot's mechanical energy output based on the task demands. While these studies serve as an important launching point to understand the mechanical behavior of foot structures, the mechanisms of energy loss within the feet of older adults are currently unclear—for example, whether the energy loss is due in part to abnormal muscle activation or strength and/or to alterations in structural properties (e.g., passive tissues or bone morphology). Furthermore, our current understanding of foot function in older adults is limited to mostly level-ground walking [6,7], and much is unknown about how older adults adapt their foot mechanics when tasked with activities that require net energy dissipation (e.g., walking downhill, stair descent, gait termination) or net energy generation (e.g., walking uphill, stair ascent, gait initiation).

In our opinion, uncovering the underlying mechanisms of the foot's mechanical energy output across a range of locomotor demands has the potential to refine our understanding of the mobility deficits facing our aging population. For example, understanding foot's energy losses could update the mechanistic knowledge of the characteristic distal-to-proximal redistribution to power locomotion [1] thought to precipitate deleterious effects on the whole-body metabolic energy cost of walking [8]. Additionally, examining how the foot's energy is lost or dissipated across a series of locomotor tasks could provide novel insights into balance control and instability, adding to the current knowledge regarding the link between foot sensation, plantar pressure distribution, and balance control [9]. As we strive to advance our theoretical knowledge of foot function in older adults, inspirations can be drawn from the latest trends in the mechanistic study of young asymptomatic feet.

## 2. Transformations in the Field of Human Foot Biomechanics

While many biomechanics studies have historically simplified the foot as a single rigid body segment [3], recent advancements in instrumentation, analytical and experimental techniques, and in vivo imaging have expanded our capabilities to quantify mechanical energy sources, at least in young asymptomatic feet. For example, multi-segment foot models integrated with non-rigid mechanics [3,10] can identify the mechanical energy utilized by all active and passive structures in various foot regions. Moreover, the integrative use of in vivo ultrasound imaging with fine-wire electromyography, combined with experiments that induce nerve blocks, have shaped our understanding of the versatility of foot structures (e.g., intrinsic muscles and tendons) to modulate the mechanical energy output based on task demands [11–13]. The latest technology involving biplane fluoroscopy has enabled simultaneous analyses of soft tissue mechanics and bone and articulating surface kinematics [14], including coupled multi-planar motions (e.g., tibiotalar, subtalar, talonavicular and calcaneocuboid joints) that are often inaccessible through traditional motion capture methods [15]. Altogether, these latest innovations have transformed our understanding of the function of young asymptomatic feet: what was once viewed as a stiff lever and a shock absorber has now been refined as a versatile machine that can seamlessly vary its

stiffness [11] and mechanical energy (dissipation, absorption and/or generation) [12,13] specific to the task demands.

While feet in young adults can act as their own source of mechanical energy to assist gait propulsion, another key function is to alter the leverage for more proximal muscles. The foot's lever function has been known for several decades [4]: specifically, its ability to modify the force-generating potential of more proximal muscles (e.g., ankle plantar flexors) by altering the contractile velocity (i.e., shifting force–velocity operating regions). Recent in vivo experiments have verified the foot's lever function and contributed to a refined understanding of how the foot affects the lower extremity mechanical and metabolic energy demands of locomotion. For example, devices that add stiffness to the foot (e.g., shoes and/or insoles) can enhance the ankle plantar flexor force production via a slower contractile velocity [16], which can afford whole-body metabolic energy savings in some locomotion tasks, such as during fast walking [16]. Moreover, foot structures have inherent mechanisms that may regulate its stiffness (and, hence, the lever function), including structural (e.g., the curvature of the transverse arch [17]) and neuromechanical (e.g., the activation of intrinsic foot muscles [11]) features. All of these studies in young adults may offer a theoretical basis for examining the foot's lever function and its consequences on locomotor performance in older adults.

### 3. Towards a Mechanistic Understanding of the Age-Related Decline in Foot Function

For researchers aiming to illuminate the mechanisms of altered foot function and their influence on mobility outcomes (e.g., gait mechanics, whole-body metabolic energy cost, balance control) in older adults, a major obstacle is unraveling the concurrent alterations in tissue morphology, structural properties and neuromechanical functions. Structurally, older adults have smaller and weaker intrinsic foot muscles and a more dissipative fat pad [18], which could potentially alter foot energetics during locomotion. Moreover, there are numerous skeletal deformities in older adults that affect joint mobility [18] (e.g., hallux valgus/rigidus and flattening of the longitudinal arch) that we speculate could alter the foot's ability to regulate stiffness and, hence, the lever function. There are also purported changes in connective tissue architecture with aging, such as the altered insertion sites of the Achilles tendon on the calcaneus [19]—which we speculate could potentially disrupt the mechanical energy transfer between the foot and the ankle. Neuromechanically, it is likely that these structural changes, combined with decreased sensory feedback [20] and altered plantar pressure distribution [9] with aging, could alter intrinsic foot muscle activations. Yet, there is currently little knowledge of the neuromechanics of the intrinsic foot muscles in older adults and how foot muscle activations affect overall mobility outcomes. In the presence of numerous structural and functional alterations, interpreting age-related changes in foot mechanics and energetics will be undoubtedly complex. We believe that these challenges are best met through collaborative efforts among researchers to leverage emerging tools and techniques to examine the independent and interdependent effects of structural changes (e.g., via non-invasive imaging) and neuromechanical adaptations (e.g., integrative modeling and experimental approaches) (Figure 1).

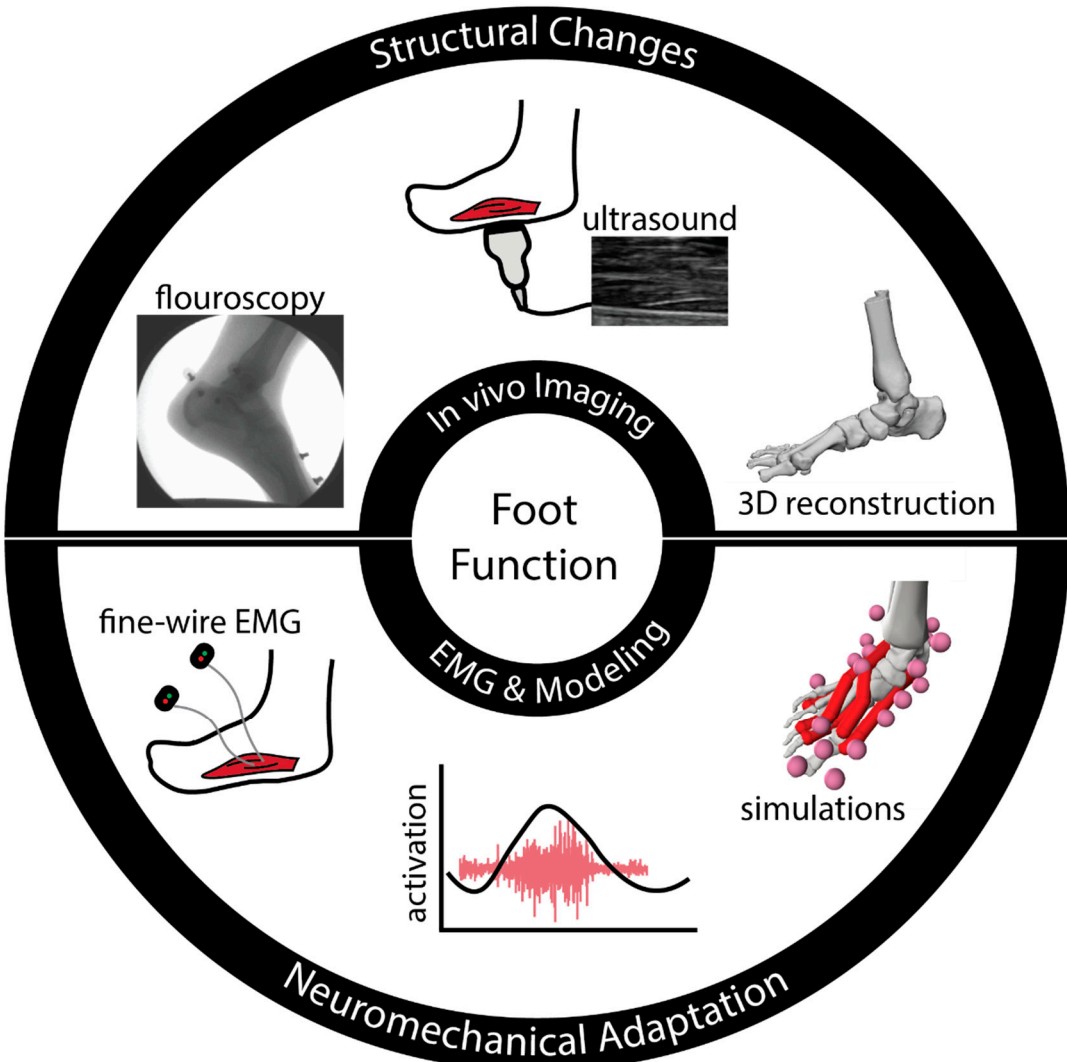

**Figure 1.** To gain a mechanistic understanding of the concurrent changes in foot structural and neuromechanical functions and their impact on mobility outcomes in older adults, future investigations may integrate contemporary approaches and tools that are currently transforming the field of human foot biomechanics.

## 4. Interventions to Rejuvenate the Foot Mechanics and Energetics in Older Adults

Mechanistic studies of feet in older adults will pave the way for future interventions aimed at targeting the structural changes and declines in neuromechanical function. To address structural issues, there are cost-effective solutions to alter the foot properties (e.g., stiffness and/or mobility of joints) and enhance leverage and mechanical energy output, such as footwear modifications [16,21]. Recent advancements in footwear designs, such as those integrated with carbon fiber materials, are revolutionizing athletic competitions such as marathons. We contend that similar footwear modifications tailored for older adults have the potential to enhance mobility outcomes. In addition to wearable devices to modify the foot structural properties, therapies that specifically target the neuromuscular function hold promise. For example, strengthening programs for intrinsic foot muscles may be promising for mitigating age-related declines in mobility outcomes [22]. There are also established techniques to enhance the sensory function, such as the use of stochastic resonance [23], which may be well-suited to improve the neuromechanical functions of intrinsic foot muscles. As foot-based interventions become more mainstream, we emphasize the importance of integrating sound and novel experimental approaches to establish cause–effect relationships to evaluate the efficacy of the interventions. For example, a

pervasive issue in the study of footwear biomechanics is the difficulty of isolating the kinetics and kinematics of the foot joints within the shoe [24]. To address this issue, new techniques may require novel model-based computations of shoe properties combined with in-shoe sensors [25] and imaging (e.g., biplane fluoroscopy). We also highlight the need for interventions that not only target the flexion/extension of mobile joints but also consider the coupled multi-planar motion of the tibiotalar joint and surrounding joints that provide a greater range of motion in the frontal and transverse planes (e.g., subtalar, talonavicular, and calcaneocuboid joints). The consideration of these mechanistic factors should pave the way for impactful and cost-effective interventions that can assist mobility in our aging population.

## 5. Concluding Remarks

As humans age, there will be natural and inevitable changes in foot structure and function. Our capacity to diagnose foot abnormalities and prescribe targeted interventions will hinge on our ability to unravel the concurrent changes in the foot structural and neuromechanical functions. In our opinion, the field is well primed to overcome these challenges if we can harness the ongoing transformation of the field of foot biomechanics and apply contemporary tools to discover novel insights for promoting healthy aging.

**Author Contributions:** Conceptualization, K.Z.T., R.L.K., A.L.L., L.A.K., M.J.R. and J.R.F.; writing-original draft preparation, K.Z.T., R.L.K., A.L.L., L.A.K., M.J.R. and J.R.F.; writing-review and editing, K.Z.T., R.L.K., A.L.L., L.A.K., M.J.R. and J.R.F.; visualization, R.L.K. All authors have read and agreed to the published version of the manuscript.

**Funding:** Research reported in this publication was supported by the National Institute of Arthritis and Musculoskeletal and Skin Diseases and the National Institute of Aging of the National Institutes of Health under Award Numbers R01AR081287 (awarded to J.R.F. and K.Z.T.), K01AR080221 (awarded to A.L.L.) and F32AG067675 (awarded to R.L.K.). The content is solely the responsibility of the authors and does not necessarily represent the official views of the National Institutes of Health.

**Institutional Review Board Statement:** Not applicable.

**Informed Consent Statement:** Not applicable.

**Data Availability Statement:** Not applicable.

**Conflicts of Interest:** The authors declare no conflict of interest.

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
