# Peer review of "Mechanics and Energetics of Human Feet: A Contemporary Perspective for Understanding Mobility Impairments in Older Adults"

_2673-7078, doi:10.3390/biomechanics2040038_

Round 1
Reviewer 1 Report
Thank you very much for considering your work to publish it with us.
As either scientific review, it would have to have:
PRISMA Guide to ensure the scientific level of the paper
CASP check list to ensure the metodological quality
-The paper doesnt follow any scientific structure, with material and methods, results...
The paper is not suitable to publish in the present form
Author Response
Thank you for reviewing our manuscript. This paper was an invited submission for a perspective review article, and not a systematic review or an original research article. As such, we feel that the PRISMA Guide or CASP checklist is not relevant for this paper. Our paper does not follow traditional scientific structures (e.g., material and methods, results, etc) and we formatted the paper based on the topics that were central to our perspective review. With the approval from the editor, we have kept the structure from our first submission.
Reviewer 2 Report
Biomechanics (MDPI) -1836150
Type of manuscript: Perspective
Title: Mechanics and energetics of human feet: a contemporary perspective for understanding mobility impairments in older adults
In general, the current study is to prospectively investigate foot biomechanics that is less being paid attention to its mechanical contributions in terms of energy consumption and neuromechanical functions declined in age-related older adults and in concern of their mobility in daily life. Specifically, the paper provides great insights into the potential improvement of mobility by further concluding remarks on the foot biomechanics related to its structure and neuromechanical functions. A few comments are provided in the following list so that this prospective paper could be enriched, specifically for the elderly who are impaired in balance and locomotion.
Minor comments
Line 32, in the 1st session of the “Current understanding of mechanics and energetics of older adults”, the older adults whose gait (foot) conditions are healthy or impaired being tested in the cited studies could have been clearly stated and mentioned throughout the entire contents of the paper, so that the foot normality of the elderly could be well distinctively and practically identified. Furthermore, in this 1st session of the contents, more elderly studies cited for balance control challenges during (perturbed) gait could have also been paid more attention to, specifically linking to the age-related decline in foot function such that the elderly mobility would be retained and restored being proposed by the prospective paper detailed in the “Transformations in the field of human foot biomechanics”, “Towards a mechanistic understanding of age-related decline in foot function” and the concluding remarks stated at the end.
Line 51~52
This paper states that “Until recently, …relatively few studies investigating foot mechanical energetics in older adults [5, 6].” The following study cited may enrich the contents of the paper.
Pol F, Baharlouei H, Taheri A, Menz HB, Forghany S. Foot and ankle biomechanics during walking in older adults: A systematic review and meta-analysis of observational studies. Gait Posture. 2021 Sep;89:14-24. doi: 10.1016/j.gaitpost.2021.06.018. Epub 2021 Jun 26. PMID: 34217001.
Line 74~76 : This paper states that “Additionally, examining how the foot’s energy is lost or dissipated …. could provide novel insights into balance control and instability that may contribute … falling.” Also, line 84~86, it states that “… multi-segment foot models … can identify the mechanical energy performed by all active and passive structures in various foot regions.” Since the energy flow lost or gained in the human foot during locomotion is biomechanically calculated and physiologically well associated with the lower limbs’ muscle contractions, some EMG studies of the elderly in gait could be well cited to strengthen the depth of the current paper.
In addition, the plantar pressure-based technology and findings can be supplemented to the contents of the paper providing further insights into the concluding remarks.
(2022) Plantar sensation, plantar pressure, and postural stability alterations and effects of visual status in older adults, Somatosensory & Motor Research, 39:1, 55-61, DOI: 10.1080/08990220.2021.1994940
Renganathan, G., Kurita, Y., Ćuković, S., Das, S. (2022). Foot Biomechanics with Emphasis on the Plantar Pressure Sensing: A Review. In: Subburaj, K., Sandhu, K., Ćuković, S. (eds) Revolutions in Product Design for Healthcare. Design Science and Innovation. Springer, Singapore. https://doi.org/10.1007/978-981-16-9455-4_7
Reviewer 3 Report
Gait analysis has been recognized as a standard and powerful tool used to capture human locomotion and quantify the related parameters. Actually new tools as smart insoles have been developed and are currently transforming the knowledge of human foot biomechanics. We recommend authors to refer to these devices in the manuscript by including some reference.
Author Response
We appreciate the thoughtful comments and suggestions by the reviewer to highlight the importance of smart insoles in the field of foot biomechanics. Reviewer #2 had similar suggestions about referencing pressure-based technology. We agree that in-shoe sensors can provide valuable insights into human foot biomechanics. We have thus added references for studies involving insoles/pressure sensors.